# Tigecycline Containing Polymethylmethacrylate Cement Against MRSA, VRE, and ESBL—In Vitro Mechanical and Microbiological Investigations

**DOI:** 10.3390/antibiotics13111102

**Published:** 2024-11-19

**Authors:** Michael Abramowicz, Andrej Trampuz, Klaus-Dieter Kühn

**Affiliations:** 1KABEG, LKH Villach, 9500 Villach, Austria; michael.abramowicz@kabeg.at; 2Faculty of Health, School of Clinical Medicine, Queensland University of Technology (QUT), Brisbane 4006, Australia; andrej.trampuz@qut.edu.au; 3Department of Orthopaedics and Trauma, Medical University Graz, 8036 Graz, Austria

**Keywords:** tigecycline, PMMA, mechanical properties, efficacy, MRSA, VRE, ESBL

## Abstract

Background: The use of antibiotic-loaded bone cements (ALBCs) in arthroplasty has been well established for the prevention and treatment of infections. Tigecycline (Tig), a broad-spectrum antibiotic, has shown efficacy against various pathogens, including vancomycin-resistant strains. Method: ISO and DIN mechanical and microbiological inhibition zone tests were performed on PMMA cement with manually added Tigecycline. Results: Manually adding 0.5 and 1.0 g Tigecycline to PMMA always meets the mechanical requirements of ISO and DIN standards. Mixtures containing 0.5 g were microbiologically effective for up to 7 days and those containing 1.0 g were effective for 28–42 days. Conclusion: In revision surgery, manually adding Tigecycline in doses of 0.5–1 g to 40 g of PMMA is effective against MRSA, VRE, and ESBL without negatively affecting the cement’s properties.

## 1. Introduction

Prosthetic joint infection (PJI) is a devasting complication after joint replacement. The treatment of joint infections is a major and demanding challenge for the patient and the surgeon [1]. In addition to the radical debridement of all septic tissues, systemic and local anti-infective agents are used to support surgery. Systemic administration of antibiotics mainly reduces the hematogenous spread of germs. Local active ingredient application is effective directly at the site of infection. ALBCs are used as local drug carriers, especially because they can facilitate the high local elution of the drug [2].

In the context of septic revisions, antibiotics are often admixed with PMMA cement [3], given that in commercially available ALBCs [4], only a limited number of antibiotics are used [5,6]. In cases of septic infection with complex etiologies [7,8], the use of locally administered antibiotics or antibiotic combinations is a necessity [9,10]. Vancomycin is frequently incorporated into the cement for the treatment of MRSA infections, as evidenced by the findings of Lee et al. (2016) and Paz et al. (2015) [11,12]. In the event that vancomycin resistance [13] is present (VRE), daptomycin or linezolid, for instance, can be considered for inclusion in the cement [14]. In the case of Gram-negative infections, colistin, meropenem, and numerous other antibiotics are employed, depending on the level of resistance [15,16]. Tigecycline (Tig), a glycylcycline derivative of tetracyclines, is a broad-spectrum antibiotic that is active against numerous Gram-positive, Gram-negative, aerobic, and anaerobic pathogens [16,17,18,19,20]. In addition, it is effective against atypical pathogens, including methicillin-resistant *Staphylococcus aureus* (MRSA), vancomycin-resistant enterococci (VRE), and extended-spectrum beta-lactamase (ESBL)-forming pathogens [21,22]. These findings have been corroborated by various studies [15,16,19,20,23,24]. Tig circumvents two significant resistance mechanisms: efflux pumps and ribosomal protection mechanisms [25]. In specific instances of vancomycin-resistant bacteria or intricate Gram-negative infections, Tigecycline may be a promising alternative [21,26].

Tig shows synergistic effects in combination with other antibiotics—like gentamicin—in the treatment of multidrug-resistant bacteria [15,22,26,27]. There is a paucity of research investigating the efficacy of Tig and PMMA cement as a local antimicrobial combination. Tig is released from PMMA cement, as evidenced by Nichol et al. (2016) and Lastinger et al. (2020) [28,29]. The mechanical properties of Tig-containing PMMA may even be influenced in low dosages to a limited extent [26,28,29,30]. For this reason, standard low and high dosages (0.5 and 1.0 g/40 g) of antibiotics were employed in the PMMA, with mechanical stability tested in accordance with ISO 5833 [31] and DIN 53435 [32]. Moreover, the effectiveness of Tig eluted from the prepared mixtures against MRSA, VRE, and ESBL was evaluated through an inhibition zone test.

## 2. Results

The Tig used was an orange-colored powder (Figure 1a). The addition of the ground powder to the PMMA powder resulted in a change in the color (Figure 1b) of the Palacos cement with and without gentamicin from its typical green to an orange-brown hue (Figure 1c).

### 2.1. ISO Compressive Strength

All tested TIG-containing cements met the ISO compression specification. The compression strength values observed in the Tig-containing groups were found to be similar, with a range of 78.50 MPa to 84.85 MPa. The standard deviation between the groups was low, with no statistical outliers. The highest values, with an average compressive strength of 84.85 MPa, were observed for Palacos R with 0.5 g Tig (Figure 2). The compression of Palacos RG with 0.5 and 1.0 g Tig is statistically indistinguishable.

In our study, for statistical reasons, we compared the compressive strength of Palacos R and Palacos R plus 0.5 g Tigecycline. The mean compressive strength values were 79.6 MPa for Palacos R and 84.85 MPa for Palacos R plus 0.5 g Tigecycline. We performed an independent *t*-test to determine if the difference in compressive strength between the two groups was statistically significant.

The results showed a statistic of 2.59 and a *p*-value of 0.019 (<0.05). Since the *p*-value is less than the conventional significance level (*p* < 0.05), we conclude that there is a statistically significant difference in the compressive strength of Palacos R and Palacos R plus 0.5 g Tigecycline, with the latter having a significantly higher compressive strength.

We also compared the compressive strength of Palacos R+G and Palacos R+G plus 0.5 g Tigecycline. The mean compressive strength values were 87.46 MPa for Palacos R+G and 84.85 MPa for Palacos R+G plus 0.5 g Tigecycline. We performed an independent *t*-test to determine if the difference in compressive strength between the two groups was statistically significant.

The results showed a statistic of 1.29 and a *p*-value of (*p* = 0.21). Since the *p* < 0.05 value is greater than the conventional significance level of 0.05, we conclude that there is no statistically significant difference in the compressive strength of Palacos R+G and Palacos R+G plus 0.5 g Tigecycline.

The *p*-value was calculated for the compressive strength of Palacos R+G with a mean value of 87.46 MPa and Palacos R+G plus 1.0 g Tigecycline with a mean value of 78.5 MPa. The *p* = 0.0036 indicates that there is a statistically significant difference between the two groups, as the *p*-value is much smaller than the conventional significance level of *p* < 0.05. This suggests that the compressive strength of Palacos R+G is significantly higher than that of Palacos R+G plus 1.0 g Tigecycline.

### 2.2. ISO Bending Strength

All Tig-containing cements fulfilled the requirements of 50 MPa. The addition of 0.5 g Tig to the references slightly decreases the bending strength, and Palacos R+G with 1 g of Tig showed the lowest bending strength just above the limit (Figure 3).

We compared the bending strength of Palacos R and Palacos R plus 0.5 g Tigecycline. The mean bending strength values were 72.3 MPa for Palacos R and 67.3 MPa for Palacos R plus 0.5 g Tigecycline. We performed an independent test to determine if the difference in bending strength between the two groups was statistically significant. The results showed a t-statistic of 2.47 and *p* = 0.024. Since the *p*-value is less than the conventional significance level of 0.05, we conclude that there is a statistically significant difference in the bending strength of Palacos R and Palacos R plus 0.5 g Tigecycline, with the former having a significantly higher bending strength.

We further compared the bending strength of Palacos R+G and Palacos R+G plus 0.5 g Tigecycline. The mean bending strength values were 65.79 MPa for Palacos R+G and 67.3 MPa for Palacos R+G plus 0.5 g Tigecycline. We performed an independent *t*-test to determine if the difference in bending strength between the two groups was statistically significant. The results showed a t-statistic of −0.75 and a *p*-value of 0.47. Since the *p*-value is greater than the conventional significance level of 0.05, we conclude that there is no statistically significant difference in the bending strength of Palacos R+G and Palacos R+G plus 0.5 g Tigecycline.

Additionally, we compared the bending strength of Palacos R+G and Palacos R+G plus 1 g Tigecycline. The mean bending strength values were 65.79 MPa for Palacos R+G and 67.3 MPa for Palacos R+G plus 1 g Tigecycline. We performed an independent *t*-test to determine if the difference in bending strength between the two groups was statistically significant. The results showed a t-statistic of −0.75 and a (*p* = 0.47). Since the *p*-value is greater than the conventional significance level of 0.05 (*p* < 0.05), we conclude that there is no statistically significant difference in the bending strength of Palacos R+G and Palacos R+G plus 1 g Tigecycline.

### 2.3. ISO Bending Modulus

The results of the bending modulus exhibited a similar range without any significant statistical outliers. The modulus of Palacos R with 0.5 g TIG and Palacos RG with 0.5 g TIG exhibited comparable values. In contrast, Palacos RG with 1.0 g TIG exhibited a slightly reduced modulus (Figure 4).

To compare the bending modulus of different formulations of Palacos, we performed independent two-sample *t*-tests. This statistical method is used to determine if there is a significant difference between the means of two independent groups.

Palacos R (mean = 2.628 MPa) vs. Palacos R + 0.5 g (mean = 2.656 MPa): T-Value: −1.084 (*p* = 0.283).

Interpretation: The *p* = 0.283 is greater than 0.05, indicating that the difference between the means of Palacos R and Palacos R + 0.5 g is not statistically significant. The T-value of −1.084 shows that the mean of Palacos R + 0.5 g is slightly higher, but this difference is not large enough to be considered significant.

Palacos R+G (mean = 2.552 MPa) vs. Palacos R+G + 0.5 g (mean = 2.624 MPa): T-Value: −2.789 (*p* = 0.007). Interpretation: The *p*-value of 0.007 is less than 0.05, indicating that the difference between the means of Palacos R+G and Palacos R+G + 0.5 g is statistically significant. The T-value of −2.789 shows that the mean of Palacos R+G + 0.5 g is significantly higher than that of Palacos R+G.

Palacos R+G (mean = 2.552 MPa) vs. Palacos R+G + 1 g (mean = 2.373 MPa): T-Value: 6.933; *p* = 3.80 times 10^−9^. Interpretation: The extremely small *p*-value (*p* = 3.80 times 10^−9^) is much less than 0.05, indicating that the difference between the means of Palacos R+G and Palacos R+G + 1 g is highly significant. The T-value of 6.933 shows that the mean of Palacos R+G is significantly higher than that of Palacos R+G + 1 g.

### 2.4. DIN Bending Strength

Despite the existence of several discrepancies between the DIN bending strength test and the ISO 5833 bending strength test, both tests yielded comparable outcomes and exhibited analogous trends. The addition of 0.5 g Tig to the references slightly decreased the DIN bending strength, and the addition of 1 g of Tig to Palacos R+G resulted in the lowest bending strength below the DIN limit of 65 MPa (Figure 5). 

### 2.5. DIN Impact Strength

All Tig-containing PMMA specimens exhibited significant divergence from the reference values. A comparison of the Palacos R reference value with the DIN impact strength of Palacos R plus 0.5 g Tig reveals a reduction from 7.5 kJ/m^2^ to 3.92 kJ/m^2^. This equates to a reduction of 63.73% in the DIN bending strength. A comparison of the reference value Palacos R+G with the admix of 0.5 g Tig reveals a reduction of 14.06% in the DIN impact strength, resulting in a value of 2.72 kJ/m^2^ (85.93%). Nevertheless, the addition of 1 g of Tig results in a further reduction in the DIN bending strength by 0.2 kJ/m^2^, bringing the total reduction to 0.4 kJ/m^2^ (12.5%) (Figure 6).

To compare the DIN bending strength of different formulations of Palacos, we performed independent two-sample *t*-tests. This statistical method is used to determine if there is a significant difference between the means of two independent groups. 

Palacos R (mean = 87.4 MPa) vs. Palacos R + 0.5 g (mean = 71.21 MPa): T-Value: 12.541; *p* = 3.69 times 10^−18^. Interpretation: The extremely small *p*-value (*p* = 3.69 times 10^−18^) is much less than 0.05, indicating that the difference between the means of Palacos R and Palacos R + 0.5 g is highly significant. The T-value of 12.541 shows that the mean of Palacos R is significantly higher than that of Palacos R + 0.5 g.

Palacos R+G (mean = 71.21 MPa) vs. Palacos R+G + 0.5 g (mean = 67.03 MPa): T-Value: 3.238; *p*-Value = 0.002. Interpretation: The *p* = 0.002 is less than *p* < 0.05, indicating that the difference between the means of Palacos R+G and Palacos R+G + 0.5 g is statistically significant. The T-value of 3.238 shows that the mean of Palacos R+G is significantly higher than that of Palacos R+G + 0.5 g.

Palacos R+G (mean = 71.21 MPa) vs. Palacos R+G + 1 g (mean = 55.78 MPa): T-Value: 11.952; *p* = 2.79 times 10^−17^. Interpretation: The extremely small *p*-value (*p* = 2.79 times 10^−17^) is much less than 0.05, indicating that the difference between the means of Palacos R+G and Palacos R+G + 1 g is highly significant. The T-value of 11.952 shows that the mean of Palacos R+G is significantly higher than that of Palacos R+G + 1 g.

Summary: The difference between Palacos R and Palacos R + 0.5 g is highly significant. The difference between Palacos R+G and Palacos R+G + 0.5 g is significant. The difference between Palacos R+G and Palacos R+G + 1 g is highly significant.

To compare the impact strength according to DIN 53435 of different formulations of Palacos, we performed independent two-sample *t*-tests. This statistical method is used to determine if there is a significant difference between the means of two independent groups.

Palacos R (mean = 7.5 kJ/m^2^) vs. Palacos R + 0.5 g (mean = 3.92 kJ/m^2^): T-Value: 13.865; *p* = 4.57 times 10^−20^. Interpretation: The extremely small *p*-value of 4.57 times 10^−20^ is much less than *p* < 0.05, indicating that the difference between the means of Palacos R and Palacos R + 0.5 g is highly significant. The T-value of 13.865 shows that the mean of Palacos R is significantly higher than that of Palacos R + 0.5 g.

Palacos R+G (mean = 3.2 kJ/m^2^) vs. Palacos R+G + 0.5 g (mean = 2.72 kJ/m^2^): T-Value: 1.859; *p* = 0.068. Interpretation: The *p* = 0.068 is greater than *p* < 0.05, indicating that the difference between the means of Palacos R+G and Palacos R+G + 0.5 g is not statistically significant. The T-value of 1.859 shows that the mean of Palacos R+G is slightly higher, but this difference is not large enough to be considered significant.

Palacos R+G (mean = 3.2 kJ/m^2^) vs. Palacos R+G + 1 g (mean = 0.26 kJ/m^2^): T-Value: 11.387; *p* = 2.03 times 10^−16^. Interpretation: The extremely small *p*-value (*p* = 2.03 times 10^−16^) is much less than *p* < 0.05, indicating that the difference between the means of Palacos R+G and Palacos R+G + 1 g is highly significant. The T-value of 11.387 shows that the mean of Palacos R+G is significantly higher than that of Palacos R+G + 1 g. Summary: The difference between Palacos R and Palacos R + 0.5 g is highly significant.

The difference between Palacos R+G and Palacos R+G + 0.5 g is not significant. The difference between Palacos R+G and Palacos R+G + 1 g is highly significant.

### 2.6. Microbiology Results

Tig was released from the PMMA cements and demonstrated inhibition zones against all the tested bacteria. Palacos R+G without Tig showed no efficacy against MRSA (Figure 7a).

### 2.7. Efficacy of Tig Against MRSA

All Tig-containing cements were found to be effective against MRSA until seven days after elution extraction. The efficacies of Tig with a dose of 0.5 g in Palacos R and Tig with the same dose in Palacos RG are essentially indistinguishable. A dose of 1.0 g Tig in the Palacos RG exhibited the greatest inhibitory effect against MRSA and demonstrated partial efficacy for up to 28 days. Nevertheless, the mean inhibition zone diameter after 28 days was 6 mm, with a high standard deviation of 5.29 mm after 14 days and a standard deviation of 0 mm after 28 d. All cement mixtures containing Tig demonstrated optimal efficacy within the first hour. Subsequently, the elution of the active ingredient and, concomitantly, the efficacy of the material decreased in a delayed manner (Figure 7b).

In our study, we compared the efficacy of Palacos R+G with 0.5 g Tigecycline and Palacos R+G with 1 g Tigecycline against methicillin-resistant *Staphylococcus aureus* (MRSA). The inhibition zones (in mm) were measured at various time points: 1 h, 1 day, 7 days, 14 days, 28 days, and 42 days.

The mean inhibition zone for Palacos R+G with 0.5 g Tigecycline was 8.28 mm, while for Palacos R+G with 1 g Tigecycline, it was 12.45 mm. We performed an independent *t*-test to determine if the difference in efficacy between the two groups was statistically significant.

The results showed a statistic of −0.76 and a *p* = 0.46. Since the *p*-value is greater than the conventional significance level of *p* < 0.05, we conclude that there is no statistically significant difference in the efficacy of Palacos R+G with 0.5 g Tigecycline and Palacos R+G with 1 g Tigecycline against MRSA.

### 2.8. Efficacy of Tig Against VRE

All Tig-containing cements demonstrated efficacy against VRE, exhibiting high elution profiles and low standard deviations over a seven-day period. Palacos R+G without Tig showed no efficacy against VRE (Figure 8a).

The microbiology results of commercial bone cement with added Tig against VRE with standard deviation are shown below. All Tig-containing cements exhibited the largest inhibition zone diameters after one hour. The most efficacious results were observed in RG with 1 g Tig at any time, with an initial average inhibition zone diameter of 25 mm, which subsequently decreased to 6.33 mm after 14 days. The standard deviation was 5.51 mm. The efficacy of R and RG with Tig against VRE exhibited only minor differences (Figure 8b).

Lastly, the specimens Palacos R + 0.5 g Tig, Palacos R+G + 0.5 Tig, and Palacos R+G +1 g Tig were also effective in inhibiting the growth of ESBL (Figure 9b), whereas Palacos R+G without Tig showed no efficacy against ESBL (Figure 9a).

In this comparison, the group with the highest concentration of Tig (Palacos R+G) demonstrated the greatest inhibition zone formation across the entire 42-day period (Figure 9b).

In the present study, the efficacy of Palacos R+G with 0.5 g Tigecycline and Palacos R+G with 1 g Tigecycline was compared against vancomycin-resistant enterococci (VRE). The inhibition zones (in mm) were measured at various time points: 1 h, 1 day, 7 days, 28 days, and 42 days.

The mean inhibition zone for Palacos R+G with 0.5 g Tigecycline was 10.6 mm, while for Palacos R+G with 1 g Tigecycline, it was 13.77 mm. An independent *t*-test was performed to ascertain whether the difference in efficacy between the two groups was statistically significant.

The results demonstrated a t-statistic of −0.50 and a (*p* = 0.63). As the *p*-value exceeded the conventional significance level of *p* < 0.05, it can be concluded that there is no statistically significant difference in the efficacy of Palacos R+G with 0.5 g Tigecycline and Palacos R+G with 1 g Tigecycline against VRE.

The most efficacious treatment against ESBL was found to be RG with 1 g Tig, with the greatest efficacy observed at the 42-day testing time point. The largest inhibition zones were observed to have a high average diameter after one hour (22 mm), which subsequently decreased to 9.67 mm after 14 days and increased again after 28 and 42 days (13.67 and 14.67 mm, respectively). The standard deviation was relatively low during the initial three time intervals, ranging from 0 to 1.15 mm, but increased during the subsequent three intervals (1.53–2.08 mm). The cements Palacos R and Palacos RG, which contain Tig with an active ingredient concentration of 0.5 g, exhibit only minor differences. Furthermore, efficacy was only observed up to the seventh day (Figure 9b).

In the present study, the efficacy of Palacos R+G with 0.5 g Tigecycline and Palacos R+G with 1 g Tigecycline was compared against extended-spectrum beta-lactamase (ESBL). The inhibition zones (in mm) were measured at various time points: 1 h, 1 day, 7 days, 14 days, 28 days, and 42 days.

The mean inhibition zone for Palacos R+G with 0.5 g Tigecycline was 10.0 mm, while for Palacos R+G with 1 g Tigecycline, it was 18.0 mm. An independent *t*-test was performed to ascertain whether the difference in efficacy between the two groups was statistically significant.

The results demonstrated a t-statistic of −1.78 and a (*p* = 0.13). As the *p*-value exceeded the conventional significance level of 0.05, it can be concluded that there is no statistically significant difference in the efficacy of Palacos R+G with 0.5 g Tigecycline and Palacos R+G with 1 g Tigecycline against ESBL.

## 3. Discussion

The results presented indicate that Tig can be easily admixed with PMMA cement [26,27,30]. Upon hardening, the cement powder assumes an intense orange color, thereby replicating the color of the Tig. To achieve a homogeneous mixture of the Tig (Tigecycline ratiopharm^®^) and PMMA, it is necessary to crush the Tig with a mortar. As with other active ingredients, it is therefore of utmost importance to ascertain which Tig powder is available in a suitable pharmaceutical form prior to its use in PMMA [33]. The incorporation of 0.5 (low) and 1.0 g (high) of Tig into 40 g of PMMA cement powder did not result in a notable alteration in the mechanical properties of the cement. All ISO and DIN tests were found to meet the requisite standards. The greatest impact on the mechanical stability was observed in the bending and impact tests. 

In a study conducted by Nichol et al. (2020) [28], the impact strength of Tig-loaded cements was found to be comparable to that of commercially available bone cements. The cement containing 10% Tig exhibited the lowest impact strength, suggesting that Tig may influence the mechanical strength of the cement. These results are comparable with our findings, although the Tig concentration was lower (1.5%) and the impact was tested with Dynstat instead of Charpy. It can be confirmed that there is a slight influence on the ISO compressive strength and the ISO bending strength [26,33]. However, the bending strength decreases significantly after the addition of 1 g Tig/40 g cement and is just within the ISO standard. It can be observed that impact and bending strength are the mechanical parameters that generally react most sensitively to the addition of active ingredients. The compressive strength and modulus of elasticity, as defined by the ISO 5833 standard, may even exhibit an increase with the addition of antimicrobial agents due to the enhanced hydrophilicity of the cement matrix [4]. Tig eluted at comparable levels from plain bone cement as well as from gentamicin-containing cement. Consequently, there is no compelling reason to advocate ALBC as a cement base for the admixing of Tig. The gentamicin did not significantly enhance the elution properties of Tig. It is often observed that gentamicin exhibits a synergistic effect when combined with other antibiotics, resulting in enhanced elution properties for both the gentamicin and the additional antibiotic [6,7,8,33]. In the absence of additional antibiotics, Tig is ineffective against *Pseudomonas* sp. and *Proteus mirabilis* [34,35]. In cases where the infection is unclear and there are gentamicin-sensitive germs present, ALBCs may be a suitable procedure, particularly given the positive results observed when 1 g Tig was admixed with 40 g Palacos R+G over a 42-day period. The synergistic effect of Tig and gentamicin essentially covers the gap that exists due to the protective effect of Tig against Gram-negative germs [21,26,34,35,36].

The optimal efficacy of Tig-containing cements can be observed within the first hour and up to day 7. Elution data revealed the presence of clinically relevant concentrations within the first hour of testing, which demonstrated antimicrobial activity up to one week later [28,34]. The concentrations of eluted Tig reached a peak around the one-hour mark and then declined, presumably due to the decomposition of the antibiotic [27,28,35]. Elgazzar et al. (2022) corroborated the toxicity of systemic application in the context of the combination of Tigecycline and gentamicin [36]. A concentration of antibiotics that is almost 1000-fold higher can be achieved without the concern of systemic side effects, whereas Tig shows excellent biocompatibility [27].

## 4. Conclusions

Tigecycline admixed with PMMA cement in doses of 0.5–1 g is effective against MRSA, VRE, and ESBL without negatively affecting the cement properties. Tigecycline-containing PMMA cement is colored orange-brown. Efficacy against MRSA, VRE, and ESBL lasts well over 7d, with Palacos R+G + 1 g Tig lasting well over 42 d.

## 5. Materials and Methods

Commercially available Palacos R and Palacos R+G cement (Heraeus Medical GmbH, Wehrheim, Germany) were used. Palacos R is a highly viscous plain PMMA cement and is characterized by the following composition: 40 g of Palacos R powder contains poly(methyl methacrylate/methacrylate), zirconium dioxide, benzoyl peroxide, and colorant E141. A 20 mL monomer liquid contains methyl methacrylate, dimethyl-p-toluidine, hydroquinone, and colorant E 141 [web Palacos R]. The highly viscous Palacos R+G is composed as follows: 40.5 g powder contains 0.5 g gentamicin (as gentamicin sulfate; https://www.heraeus-medical.com/en-us/search/?searchTerm=Instructions+for+user, accessed on 14 August 2024) [37].

Other ingredients include poly(methyl methacrylate/methacrylate), zirconium dioxide, benzoyl peroxide, and colorant E 141. A 20 mL monomer liquid contains methyl methacrylate, dimethyl-p-toluidine, hydroquinone, and colorant E 141 (web Palacos R+G). Further material characterization of Palacos R as well as Palacos R+G has been previously reported; URL: https://www.heraeus-medical.com/en-us/search/?searchTerm=Instructions+for+user accessed on 14 August 2024 [37].

The manual addition of Tig powder to PMMA powder was performed in fractions according to the recommendation of the Swiss Society for Infectious Disease [2].

The solution was prepared using ampules of Tigecycline-ratiopharm^®^ 50 mg, which contain a powder for producing solutions for infusions. Each ampule of the pharmaceutical product “Tigecyclin-ratiopharm^®^ 50 mg” contained 50 mg of tigecycline in 100 mg of powder. However, the powder within the ampules had solidified into an orange mass. Consequently, the mass was extracted from the ampule and pulverized with a mortar and pestle in order to reconstitute it as a powder. Following the grinding of the antibiotic into a cement powder, 0.5 g (equivalent to 10 ampules = low dosage) and 1 g (equivalent to 20 ampules—high dosage) of Tig were added to 40 g Palacos^®^ R and 40.5 g Palacos^®^ R+G cement powder. An overview of all Tig-containing PMMA cements and the tested parameters were summarized in Table 1. All cements were mechanically tested according to ISO 5833 and DIN 53435 [31,32].

### 5.1. ISO 5833 Compressive Strength

The ISO compressive strength test is employed to ascertain the pressure or compressive force required for the cement to lose its stability and fail. All cement rods (height: 12 mm +/−1; diameter: 6 mm +/−0, 1; 12 for each combination) were placed in the middle of a test machine capable of applying and measuring compressive force (Zwick/Roell) while running on the testXpert II Zwick/Roell software. The machine applied an increasing force until the test rod fractured, or until the 2% offset load or upper yield-point load was reached. Upon reaching one of the aforementioned endpoints, the internal pressure was measured in MPa, and the software proceeded to automatically calculate the average compressive strength and standard deviation of all tested cement rods. In order to comply with the standards set forth in ISO 5833 (2002), the rods were required to reach a minimum internal pressure of 70 MPa on average.

### 5.2. ISO 5833 Bending Modulus and Bending Strength

The test was employed to ascertain the ALBC’s resistance to bending. For each cement combination, six rectangular cement bodies (75 × 10 × 3.3 mm) were utilized. Subsequently, the cement bodies were extracted from the stainless-steel molds and placed in a four-point test rig (Zwick/Roell) running on the TestXpert II Zwick/Roell software. In order to ensure the most accurate results, great care was taken to place the test specimens as centrally as possible on the device. Subsequently, the bend test machine applied a force to the test specimen, measuring its deflection until the specimen broke. The machine software then calculated the bending and strength modulus with the average and standard deviation of each cement combination in mPa. In order to comply with the ISO standards, the specimens were required to reach a minimum bending modulus of 1800 mPa and a minimum bending strength of 50 mPa.

### 5.3. DIN Bending and DIN Impact Strength acc. to DIN 53435

In each instance, eight test bodies (length: 15 mm; width: 10 mm; depth: 3.3 mm) were utilized. To ascertain the DIN bending strength, the test bodies were placed within the DIN bending test apparatus. The apparatus commenced rotation at 100°/min, applying a bending moment of 400 Ncm to the test body. Upon the test body’s failure, the apparatus was halted, and the drag indicator displayed the bending moment of the test body at failure (Ncm). To determine the impact strength, the test bodies were placed in the DIN strength apparatus, with the pendulum placed in its starting position. Upon release of the pendulum, it collided with the test body with an impact energy of 0.5 J. The requisite impact energy (J) to fracture the test body was then displayed by the drag indicator.

As with the bending strength, the average impact strength of the eight test bodies and the standard deviation were calculated. To comply with the DIN specification, the specimens were required to reach a minimum of 65 MPa for DIN bending strength. The results of the DIN impact strength tests were compared with the reference data.

In accordance with the standard specifications, all mechanical tests are conducted with the requisite number of samples and statistical evaluation.

### 5.4. Microbiological Properties

All microbiological inhibition zone tests were conducted at the Charité University in Berlin (Proimplant Foundation), Germany. 

### 5.5. Bacterial Strains

All test bodies (25 × 10 mm) were subjected to examination at Charité, Berlin, with the objective of determining the antibacterial efficacy of inhibition zone assays against MRSA (strain *ATCC 43300*), VRE (strain *Van A*), and ESBL (strain *BJ HDL-1*). For all tested ATCC strains used in the study, the MICs and MBCs are known and documented at Charité in Berlin, Germany.

A total of 9 test bodies were tested against each bacterium, with 3 test bodies per group. The eluates of each group were employed in the tests, with a total of 3 samples per group.

### 5.6. Medium Preparation

To perform the microbiological tests, two different mediums had to be prepared. The phosphate-buffered saline (PBS) solution was employed as a buffer to extract the antibiotics from the test bodies and spacers. The Müller–Hinton Agar (MHA) medium was used to cultivate the bacterial colonies and execute the inhibition zone assays. 

To prepare one liter of phosphate-buffered saline (PBS), we utilized 10 PBS tablets manufactured by VWR Chemicals (lot number: 18K1556345) containing 137 mM sodium chloride, 2.7 mM potassium chloride, and 10 mM phosphate buffer. These were placed in a glass bottle and 1000 mL of distilled water (ddH2O) was added. Once the tablets had dissolved in the water, the lid was gently screwed onto the glass bottle in such a way that it was not sealed completely. The bottle was then autoclaved for three hours. Once the solution had been autoclaved, the lids were screwed on tightly and the bottle was allowed to cool.

To prepare 400 mL of MHA (equivalent to 20 Petri dishes), we combined 8.4 g of MH broth (OXIOD), 6 g of agar (Sigma-Aldrich), and 400 mL of ddH2O in a glass bottle. Once more, the lids of the bottles were lightly screwed on and the contents autoclaved for a period of three hours. Once the lids had been secured, the liquid agar was allowed to cool in a water bath maintained at 50 °C. Once the agar had reached 50 °C, 20 mL of the agar was pipetted into Petri dishes and left to solidify at room temperature (25 °C).

### 5.7. Test Body Preparation and Eluate Extraction

To extract the antibiotics from the cement, each of the three test bodies (triplicates) from every combination was placed in a separate 50 mL Falcon^®^ tube containing 20 mL of PBS. The tubes were then sealed, the lids covered with Parafilm^®^ to prevent leakage, and placed upside down so that the bodies were completely submerged in the PBS.

The tubes were incubated at room temperature (25 °C) for 1 h, 1 d, 7 days, 14 days, 28 days, and 42 days (Figure 10). After each time interval, 2 mL of the PBS (eluate) was removed and stored in a marked (i.e., A1, group, and cement body number) Eppendorf^®^ Tube. The rest of the PBS was discarded, and the Falcon^®^ Tubes were refilled with 20 mL of fresh PBS. The tubes were then sealed again and placed upside down until the next extraction interval.

### 5.8. Bacteria Preparation

In order to obtain bacterial suspensions for the purpose of conducting standardized microbial testing in accordance with the subsequent inhibition zone assays, the beads comprising the bacterial strains in question were plated and diluted on MHA in a Petri dish. The cultures were then incubated at 37 °C overnight.

A single colony or several colonies were removed with a swab and mixed into a saline solution (0.85% NaCl) until a McFarland standard of 0.5 (+/− 0.1) was achieved.

Inhibition zone essay: All inhibition zone assays were conducted on MHA (n = 3). For each tested strain and time interval, a new MHA plate was employed. The plates contained a special microbiological culture medium that is used to test antibiotic resistance (testing according to EUCAST). A 6 mm diameter hole was punched out in the center of the agar plate using a Pasteur pipette, into which 50 μL of the test eluates was pipetted. Each sample was placed in three agar plates. The Petri dishes were incubated overnight at 37 °C, after which the diameter of the inhibition zones was measured and documented in mm.

Gentamicin is not effective against MRSA, VRE, and ESBL (gentamicin sulfate: https://en.wikipedia.org/wiki/Gentamicin, accessed on 21 September 2024). In preparation for the microbiological tests, agar plates were prepared with Palacos R with and without Tig and Palacos RG with and without Tig in order to demonstrate that the PMMA cements used for the study had no effect on the germs tested (Figure 11).

### 5.9. Statistical Evaluation

We used independent two-sample *t*-tests to compare the means of different formulations of Palacos. This method helps determine if there is a statistically significant difference between the means of two independent groups. Here are the key steps: Data Collection: Gather means, standard deviations, and sample sizes for each group. Calculation: T-Value: Measures the difference between group means relative to the variability within the groups. *p*-Value: Indicates the probability that the observed difference is due to chance. A *p*-value less than 0.05 typically indicates statistical significance.

Software Used: The calculations were performed using Python with the SciPy library, which is widely used for scientific and statistical computing.

## Figures and Tables

**Figure 1 antibiotics-13-01102-f001:**
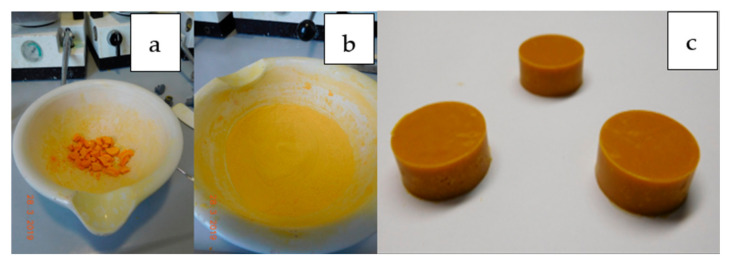
Tig preparation for bone cement addition. Left = before grinding (**a**); middle = after grinding (**b**); Tig-containing cement test bodies (**c**).

**Figure 2 antibiotics-13-01102-f002:**
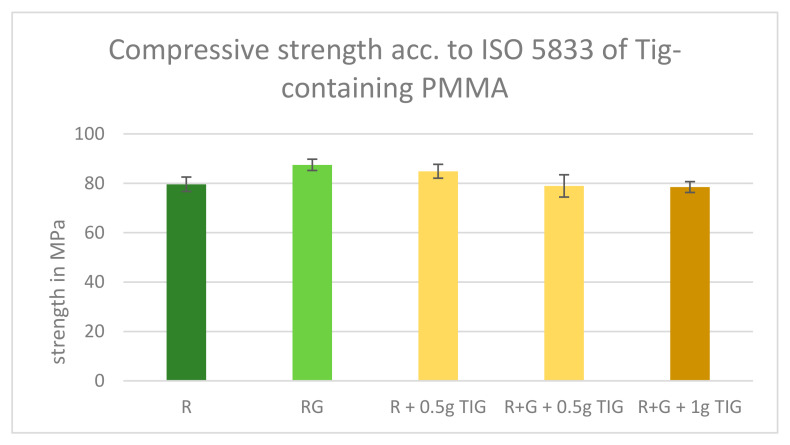
ISO compression strength results of Tig-containing PMMA. All tested cements fulfilled the requirements of 70 MPa. Legend: Palacos R—dark green; Palacos R+G—light green; orange bars = references with added Tig. Values are given as means with their corresponding standard deviation (+/−) in MPa.

**Figure 3 antibiotics-13-01102-f003:**
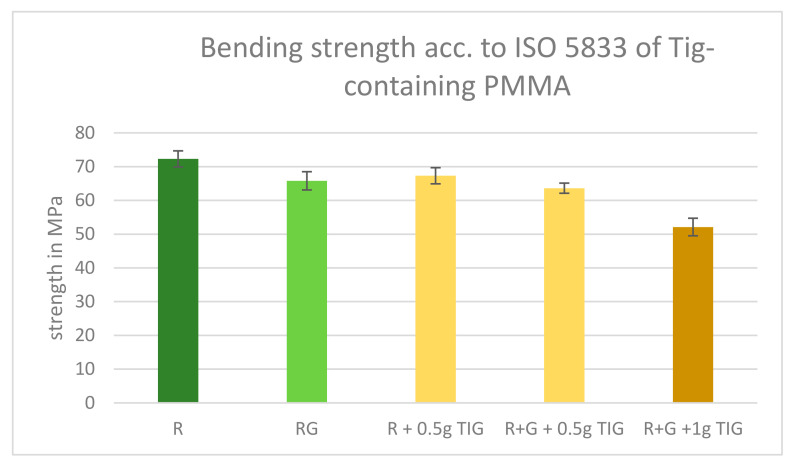
ISO bending strength results of Tig-containing PMMA. All tested cements fulfilled the required 50 MPa. Legend: Palacos R—dark green; Palacos R+G—light green; orange bars = references with added Tig. Values are given as means with their corresponding standard deviation (+/−) in MPa.

**Figure 4 antibiotics-13-01102-f004:**
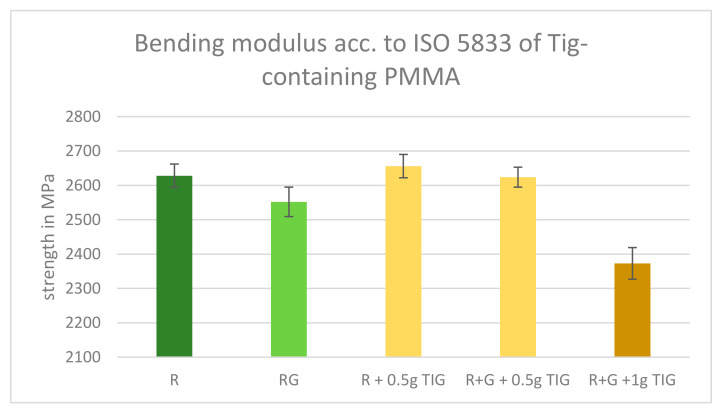
ISO bending modulus results of Tig-containing PMMA. All tested cements fulfilled the required bending modulus of 1800 MPa. Legend: Palacos R—dark green; Palacos R+G—light green; orange bars = references with added Tig. Values are given as means with their corresponding standard deviation (+/−) in MPa.

**Figure 5 antibiotics-13-01102-f005:**
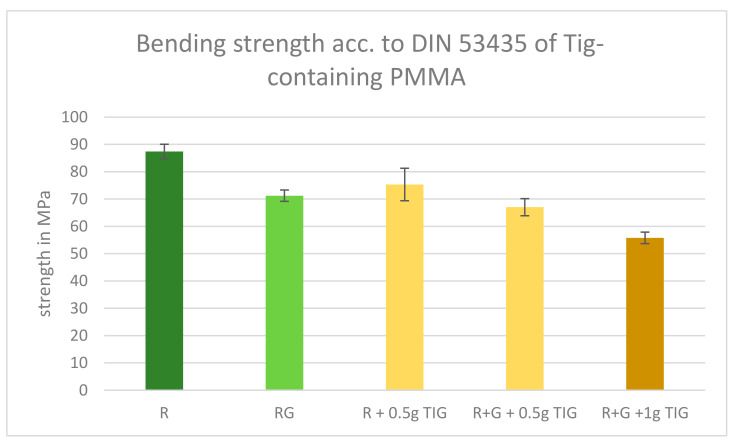
DIN bending strength results of Tig-containing PMMA. Limit: 65 MPa. Legend: Palacos R—dark green; Palacos R+G—light green; orange bars = references with added Tig. Values are given as means with their corresponding standard deviation (+/−) in MPa.

**Figure 6 antibiotics-13-01102-f006:**
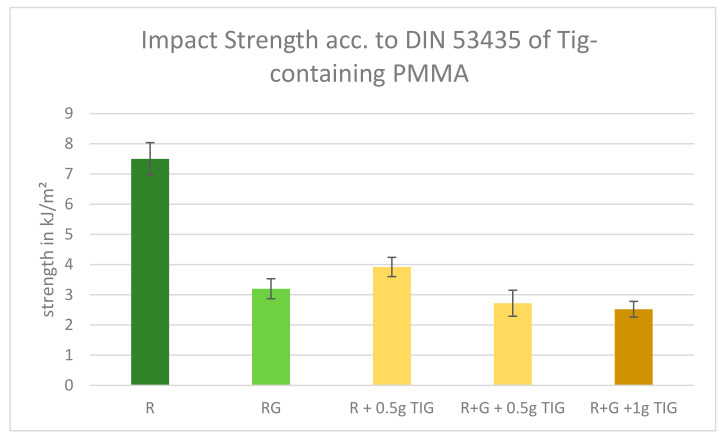
DIN impact strength results of Tig-containing PMMA. Legend: Palacos R—dark green; Palacos R+G—light green; orange bars = references with added Tig. Values are given as means with their corresponding standard deviation kJ/m^2^.

**Figure 7 antibiotics-13-01102-f007:**
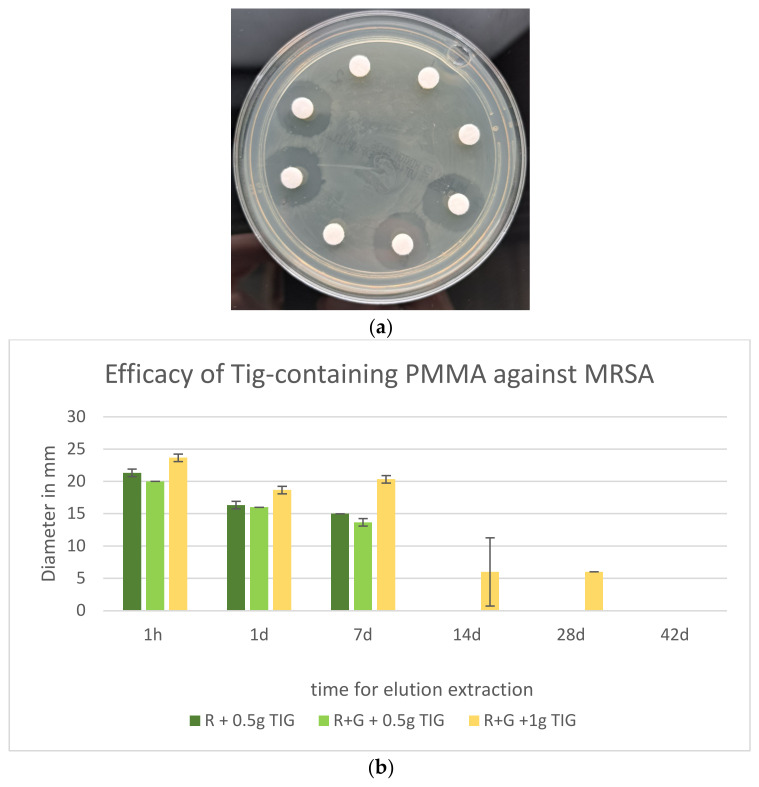
(**a**) Palacos R+G inhibition zones against MRSA. Eluates with and without Tig. Gentamicin-containing eluates without IHZ demonstrating ineffectiveness against MRSA. (**b**) Efficacy (inhibition zones in mm on MHA, 60 µL eluate) of Tig-containing Palacos R and Palacos R+G against MRSA (*n* = 3).

**Figure 8 antibiotics-13-01102-f008:**
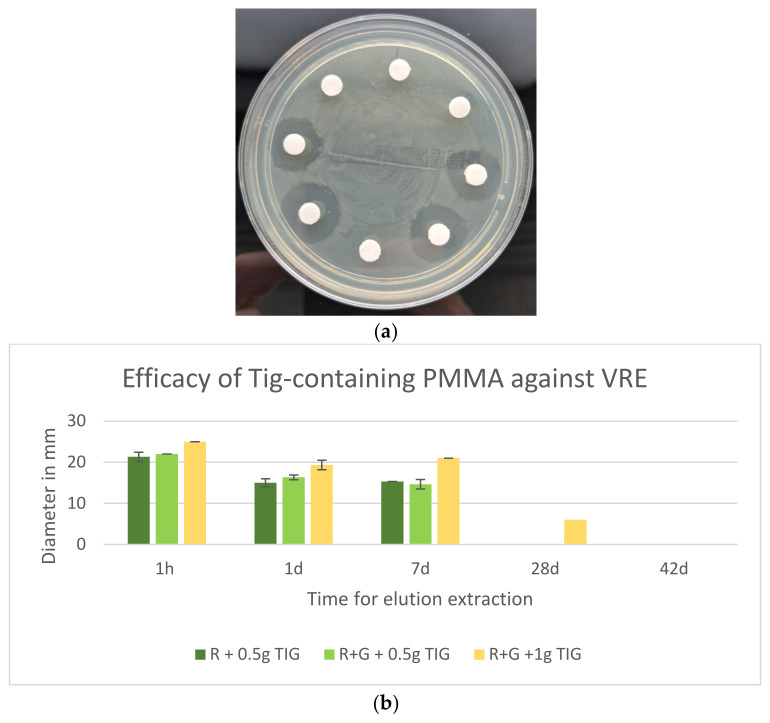
(**a**) Palacos R+G inhibition zones against VRE. Eluates with and without Tig. Gentamicin-containing eluates without IHZ demonstrating ineffectiveness against VRE. (**b**) Efficacy (inhibition zones in mm on MHA, 60 µL eluate) of Tig-containing Palacos R and Palacos R+G against VRE (*n* = 3).

**Figure 9 antibiotics-13-01102-f009:**
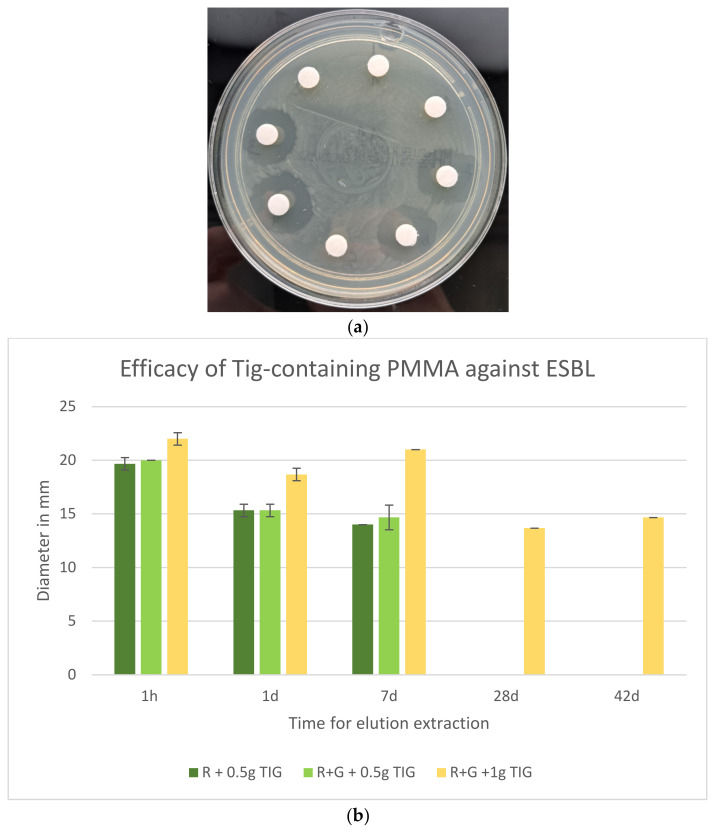
(**a**) Palacos R+G inhibition zones against ESBL. Eluates with and without Tig. Gentamicin-containing eluates without IHZ demonstrating ineffectiveness against ESBL. (**b**) Efficacy (inhibition zones in mm on MHA, 60 µL eluate) of Tig-containing Palacos R and Palacos R+G against ESBL (*n* = 3).

**Figure 10 antibiotics-13-01102-f010:**
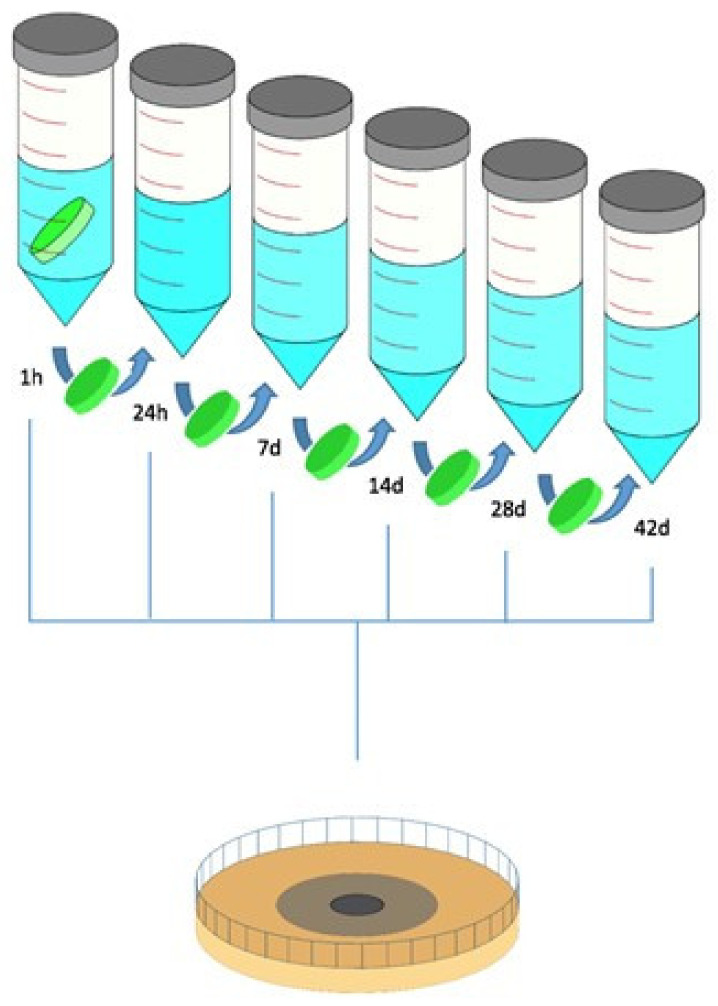
Schematic visualization of the production of the eluates tested in the inhibition zone test over a period of 42 days. The cement bodies were incubated in 20 mL 1×PBS until the indicated time points. After each time point, the bodies were moved to a fresh tube and the previous eluates were used to run the inhibition zone assay.

**Figure 11 antibiotics-13-01102-f011:**
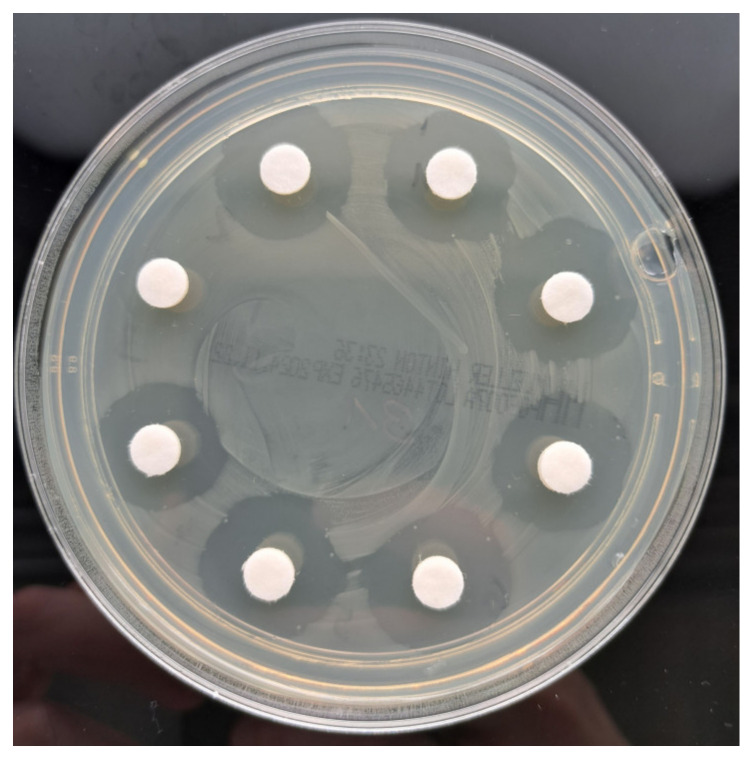
Palacos R+G inhibition zones against VRE. Eluates without Tigecycline with no inhibition zone. Palacos R+G with Tigecycline with inhibition zones against VRE.

**Table 1 antibiotics-13-01102-t001:** Testing overview: ISO 5833, DIN 53435, inhibition zone test. The dose–response and mechanical stability of the PMMA–Tigecycline mixture were the target criteria + tested; MRSA = methicillin-resistant *Staphylococcus aureus*; VRE = vancomycin-resistant *Enterococcus*; ESBL = extended-spectrum beta-lactamase; Tig = Tigecycline; reference R = Palacos R without antibiotics; reference R+G = Palacos R+G with 0.5 g gentamicin/40 g.

	MRSA	VRE	ESBL	ISO Compression	ISO Bending	ISO Modulus	DIN Bending	DIN Impact

Reference R	+	+	+	+	+	+	+	+
Reference R+G	+	+	+	+	+	+	+	+

R + 0.5 g Tig	+	+	+	+	+	+	+	+

R+G + 0.5 g Tig	+	+	+	+	+	+	+	+

R+G + 1 g Tig	+	+	+	+	+	+	+	+


## Data Availability

The original contributions presented in the study are included in the article; further inquiries can be directed to the corresponding author.

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
