# Peer review of "Tigecycline Containing Polymethylmethacrylate Cement Against MRSA, VRE, and ESBL—In Vitro Mechanical and Microbiological Investigations"

_antibiotics, 2024, doi:10.3390/antibiotics13111102_

Round 1

Reviewer 1 Report

Comments and Suggestions for Authors

I am delighted to have had the opportunity to review this article. I want to extend my gratitude to the authors for their valuable contribution. In this research, the authors investigated the use of tigecycline-loaded PMMA cement as a potential antimicrobial solution for infections caused by resistant bacteria such as MRSA, VRE, and ESBL. They examined both the mechanical properties and antimicrobial efficacy of the cement, finding that it maintained the required mechanical strength while demonstrating significant inhibition of these bacteria over an extended period. This manuscript is well-written and presents solid findings, making it a valuable addition to the field.

Author Response

Thank you very much for your effort and your kind comments

Reviewer 2 Report

Comments and Suggestions for Authors

The manuscript is well written and the conclusions are well supported by the data.  The following suggestions would improve the presentation and understanding by the audience

1. The authors should expand the introduction, adding a paragraph about the clinical impacts of infections and the role of bone cements in potentially mitigating these.  It is well understood by those in the field, but the journal has a very broad readership from many areas of antibiotics.

2. the definitions of (R), (R+G) used throughout the manuscript are not defined in the text.  This should be at the beginning of the results section (P2)

3. The authors should discuss and provide references for the expected performance of the R+G formulation, without Tig, in the inhibition zone experiments.  The readers should be able to compare the baseline efficacy of the gentamicin component in the cement.

4. The authors should give more details about the inhibition zone assay.  The methods section states "60 µl of the eluates from each 339 group per time interval were then pipetted into the corresponding wells".  But there is no description of the size/depth/placement of the wells on the petri dish, nor was there a description of the seeding of the plate with bacteria.  These details are important when comparing studies.

5.  The authors should consider removing table 1, as I am unclear to the benefit it adds.

6. The authors should include images of the zone of inhibition plates as supplementary material.

7. the authors use the convention 0,5 rather than 0.5 for decimal places in the figures.  I am not sure if the journal has a requirement, but if so it should be changed in proofing.

8. an overall proofreading for minor typos would be beneficial.  E.g. "essay" instead of "assay" (line 337), missing reference (line 264, line 252), "Gram" not capitalized (line 44) etc.

Author Response

Reviewer 2

The manuscript is well written and the conclusions are well supported by the data.  The following suggestions would improve the presentation and understanding by the audience

  1. The authors should expand the introduction, adding a paragraph about the clinical impacts of infections and the role of bone cements in potentially mitigating these. It is well understood by those in the field, but the journal has a very broad readership from many areas of antibiotics.

 Prosthetic joint infection (PJI) is a devasting complication after joint replacement .The treatment of joint infections is a major and demanding challenge for the patient and the surgeon. In addition to radical debridement of all septic tissues, systemic and local anti-infective agents are used to support surgery therapy. Systemic administration of antibiotics mainly reduces hematogenously spreading germs. Local active ingredient application is effective directly at the site of infection. ALBCs are used as local drug carriers, especially because of the high local elution of the drug During septic revision surgeries, antibiotics are manually added to the ALBCs in order to locally combat the germs relevant to the infection as effectively as possible (Borens and Ochsner 2014 as literature no 2).

  1. the definitions of (R), (R+G) used throughout the manuscript are not defined in the text. This should be at the beginning of the results section (P2)

Commercially available Palacos R and Palacos R+G cement (Heraeus Medical GmbH,Wehrheim were used. Palacos R is a high viscous plain PMMA cement and is characterized by the following composition: 40 g of Palaoc R powder contains Poly(methyl methacrylate/methacrylate), Zirconium dioxide, Benzoyl peroxide, and colorant E141. A 20 mL monomer liquid contains: methyl methacrylate, dimethyl-p-toluidine, hydroquinone, and colorant E 141 [web Palacos R]. The high visvous Palacos R+G is composed as follows: 40,5 g powder contains 0.5 g gentamicin (as gentamicin sulfate.

Other ingredients include poly(methyl methacrylate/methacrylate), zirconium dioxide, benzoyl peroxide, and colorant E 141. A 20 mL monomer liquid contains methyl methacrylate, dimethyl-p-toluidine, hydroquinone, and colorant E 141 (web Palacos R+G). Further material characterization of Palacos R as well as Palacos R+G has been previously reported [web Palacos].

The manual addition of Tig powder to PMMA powder is done in fractions according to the recommendation of the Swiss Society for Infectious Disease (Borens and Ochsner 2014).

  1. The authors should discuss and provide references for the expected performance of the R+G formulation, without Tig, in the inhibition zone experiments. The readers should be able to compare the baseline efficacy of the gentamicin component in the cement.

We have also included some corresponding images

  1. The authors should give more details about the inhibition zone assay. The methods section states "60 µl of the eluates from each 339 group per time interval were then pipetted into the corresponding wells".  But there is no description of the size/depth/placement of the wells on the petri dish, nor was there a description of the seeding of the plate with bacteria.  These details are important when comparing studies.

To perform the agar diffusion test, a 0.5 bacterial suspension was prepared based on McFarland turbidity standards. This corresponds to a cell count of 1.5 x 108 bacteria per ml. This bacterial suspension was applied to so-called “Müller-Hinton agar plates” using sterile cotton swabs, with a rotary table ensuring even distribution. The plates are a special microbiological culture medium that is used to test antibiotic resistance (testing according to EUCAST). A 6 mm diameter hole was punched out in the center of the agar plate using a Pasteur pipette, into which 50 μl of the test eluates were pipetted. Each sample was placed in three agar plates (determined in triplicate).

Gentamicin is not effective against MRSA, VRE and ESBL (gentamicin sulfate: https.//en.wikipedia.org)

In preparation for the microbiological tests, agar plates were prepared with Palacos R with and without Tig and Palacos RG with and without Tig in order to demonstrate that the PMMA cements used for the study had no effect on the germs tested.

After 24 hours of incubation at 37° Celsius in the incubator, the size of the inhibition zones was measured. The diameter of each agar plate was measured twice (perpendicular to each other) and the average of the two values ​​was calculated. The information was given in millimeters (mm).

  1. The authors should consider removing table 1, as I am unclear to the benefit it adds.

The addition of antibiotics to PMMA powder during septic revisions in endosurgery is carried out directly in the operating theatre. The surgeon thereby becomes the manufacturer of a medical device and assumes all liability. Discoloration of the cement powder is often a sign of interactions between the antibiotics used and components of the PMMA cement. For this reason, it is important for surgeons to know that when tigecycline is added to PMMA, the cement changes color significantly and this discoloration does not represent a chemical or physical reaction.

We would therefore like to ask that you do not remove this representation from the article.

  1. The authors should include images of the zone of inhibition plates as supplementary material.

See 3: we have include the images of IZT from the material to demonstrate that Genta in Palacos R+G has no efficacy against MRSA, VRE and ESBL. The images include showed the efficacy of Tig containing PMMA and Palacos R or Palacos RG was the references.

  1. the authors use the convention 0,5 rather than 0.5 for decimal places in the figures. I am not sure if the journal has a requirement, but if so it should be changed in proofing.

changed

  1. an overall proofreading for minor typos would be beneficial. E.g. "essay" instead of "assay" (line 337), missing reference (line 264, line 252), "Gram" not capitalized (line 44) etc.

changed

Reviewer 3 Report

Comments and Suggestions for Authors

The research contains the bone cement containing Tigecycline. The manuscript has enough data but I can suggest some improvements as follows

- Line 21, 22: The conclusion is not a conclusion, please consider rephrasing.

- Line 26: ALBC is used as Antibiotic Loaded Bone Cements in the text, not for Anbiotic-loaded polymethyl methacrylate bone cement; please consider using it.

- Figure 1: The purpose of the figure used is not clear; please consider renewing. 

- Figure 2-7: please use axes names for clarity; the axes names give the readers more clarity. 

- Figure 2: Please start from 0; if you want to indicate the high levels, please use figures indicating 0-50 and 50-70 graphs.

- Please give statistical data and p values for the figures

- Conclusion, please rephrase the conclusion part; in this form, it is a summary of the results.

- The powder form contains 50 mg of other ingredients. Did the Authors take this into account in the calculations?

- Line 252, 264, 278 empty brackets?

- Line 242: What is the number of samples for the experiments? 

Author Response

Reviewer 3

The research contains the bone cement containing Tigecycline. The manuscript has enough data but I can suggest some improvements as follows

- Line 21, 22: The conclusion is not a conclusion, please consider rephrasing.

In revision surgery manually adding of Tigecycline in doses of 0.5-1g to 40g of PMMA is effective against MRSA, VRE and ESBL without negatively affecting the cement properties

- Line 26: ALBC is used as Antibiotic Loaded Bone Cements in the text, not for Anbiotic-loaded polymethyl methacrylate bone cement; please consider using it.

changed

- Figure 1: The purpose of the figure used is not clear; please consider renewing.

The addition of antibiotics to PMMA powder during septic revisions in endosurgery is carried out directly in the operating theatre. The surgeon thereby becomes the manufacturer of a medical device and assumes all liability. Discoloration of the cement powder is often a sign of interactions between the antibiotics used and components of the PMMA cement. For this reason, it is important for surgeons to know that when tigecycline is added to PMMA, the cement changes color significantly and this discoloration does not represent a chemical or physical reaction.

We would therefore like to ask that you do not remove this representation from the article.

- Figure 2-7: please use axes names for clarity; the axes names give the readers more clarity.

changed

- Figure 2: Please start from 0; if you want to indicate the high levels, please use figures indicating 0-50 and 50-70 graphs.

changed

- Please give statistical data and p values for the figures

With regard to the presentation, we adhered to the specifications of ISO 5833 and DIN 53435 and did not explicitly present their information again in the article. All values ​​determined were within the standard specifications; the standard deviations are given. The values ​​shown in the figures represent mean values ​​of the individual values ​​specified by the standards including SD.

We used independent two-sample t-tests to compare the means of different formulations of Palacos. This method helps determine if there is a statistically significant difference between the means of two independent groups. (Data Collection: Gather means, standard deviations, and sample sizes for each group. Software used :The calculations were performed using Python with the SciPy library, which is widely used for scientific and statistical computing.

- Conclusion, please rephrase the conclusion part; in this form, it is a summary of the results.

changed

In revision surgery manually adding of Tigecycline in doses of 0.5-1g to 40g of PMMA is effective against MRSA, VRE and ESBL without negatively affecting the cement properties

- The powder form contains 50 mg of other ingredients. Did the Authors take this into account in the calculations?

Thank you very much for this very good and important note.

Many antibiotics are not available in pure, sterile form. Minor “contamination” cannot be avoided due to the manufacturer and is reported in pharmacopoeias. Such “impurities” are only important to the user if they significantly reduce the proportion of active substance in the antibiotic or in using drug formulations. In this case, the weight of the active substance must be adjusted accordingly. This is not the case with tigecycline.

- Line 252, 264, 278 empty brackets?

changed

- Line 242: What is the number of samples for the experiments?

Microbiological tests n=3

Reviewer 4 Report

Comments and Suggestions for Authors

The manuscript is based on the in vitro mechanical and microbiological evaluation of the commercial bone cement (plain and Gentamycin loaded Palacos) co-loaded with Tigecyclin (Tig) for potential use against antibiotic resistant species. For this purpose, commercial pharmaceutic Tigecyclin-ratiopharm® containing 50mg Tig in 100mg was re-powdered mechanically. Although the research plan seemed to be designed properly, there are important problems with the article. These are summarized below;

-          The abstract does not seem to be descriptive, too long background information very short methods and results information that does not provide a satisfactory perspective to the research conducted for this article.

-          The materials and methods sections do not provide the information on how PMMA cement had been prepared and cured, which should be provided even though it is a well known procedure.

-          There are no evaluations on the particulate content of the cement, effect of the size and homogeneous mixing of powder containing Tig in PMMA matrix.

-          For the evaluation of mechanical tests on specimens, SEM is an important tool used to describe the propagation and/or shielding of stress on the fractured surface together with the particulate distribution within the matrix. The lack of SEM analyses is an important deficiency in this article.

-          Although statistical analyses mentioned many times throughout the article, there is no information on how these analyses carried out and also p values of the comments on the results.

-          Only inhibition zones were evaluated in order to determine the efficacy of the Tig released but there is no indication on the determination of the amount of Tig released from the samples.

Author Response

Reviewer 4

The manuscript is based on the in vitro mechanical and microbiological evaluation of the commercial bone cement (plain and Gentamycin loaded Palacos) co-loaded with Tigecyclin (Tig) for potential use against antibiotic resistant species. For this purpose, commercial pharmaceutic Tigecyclin-ratiopharm® containing 50mg Tig in 100mg was re-powdered mechanically. Although the research plan seemed to be designed properly, there are important problems with the article. These are summarized below;

-          The abstract does not seem to be descriptive, too long background information very short methods and results information that does not provide a satisfactory perspective to the research conducted for this article.

Changed accordingly

-          The materials and methods sections do not provide the information on how PMMA cement had been prepared and cured, which should be provided even though it is a well known procedure.

Commercially available Palacos R and Palacos R+G cement (Heraeus Medical GmbH,Wehrheim were used. Palacos R is a high viscous plain PMMA cement and is characterized by the following composition: 40 g of Palacos R powder contains Poly(methyl methacrylate/methacrylate), Zirconium dioxide, Benzoyl peroxide, and colorant E141. A 20 mL monomer liquid contains: methyl methacrylate, dimethyl-p-toluidine, hydroquinone, and colorant E 141 [web Palacos R]. The high viscous Palacos R+G is composed as follows: 40,5 g powder contains 0.5 g gentamicin (as gentamicin sulfate. Other ingredients include poly(methyl methacrylate/methacrylate), zirconium dioxide, benzoyl peroxide, and colorant

E 141. A 20 mL monomer liquid contains methyl methacrylate, dimethyl-p-toluidine,

hydroquinone, and colorant E 141 (web Palacos R+G). Further material and instruction for use for Palacos R as well as Palacos R+G has been previously reported [web Palacos].

-          There are no evaluations on the particulate content of the cement, effect of the size and homogeneous mixing of powder containing Tig in PMMA matrix.

Thank you very much for the critical note. Palacos has been available on the market since 1958 and is considered the gold standard in cemented endoprosthetics. We have included the composition and product properties in the materials and methods section and referred to the manufacturer's corresponding websites. Please allow us the following information: With this article we want to support the surgeon in the operating theatre so that he can help the patient to survive in the event of a serious infection caused by MRSA, VRE and ESBL. For this purpose, the surgeon uses - under his own responsibility - two approved and completely known pharmaceutical or medical devices.  The surgeon makes the mixture of both products himself in the operating theatre and applies this cement directly into the bone. The surgeon follows strictly the manufacturer's instructions for mixing and applying the cement and for adding antibiotics to the cement. We have shown that if all of these instructions are followed, the surgeon will have a safe combination product to use.

We were not able to investigate whether different tigecycline qualities with different compositions and particle size distributions have an influence on the cement properties. Of course, we don't know which tigecycline the surgeon receives from his hospital pharmacy. We were unable to find any negative influence on the properties of the tigecycline from rationpharm that we used when 0.5-1.0 g of tigecycline was added to the Palacos. This statement is essential from the surgeon's perspective.

-          For the evaluation of mechanical tests on specimens, SEM is an important tool used to describe the propagation and/or shielding of stress on the fractured surface together with the particulate distribution within the matrix. The lack of SEM analyses is an important deficiency in this article.

Undoubtedly, an SEM evaluation of the cement samples can provide clues to the causes of failures in the mechanical tests. These are usually inhomogeneities or large air bubbles that arise when the cement components are mixed and can weaken the cement matrix. The ISO and DIN standards do not require any additional SEM examination of the molded bodies produced. We just wanted to show that the cement mixtures we produce with different tigecycline contents meet the requirements of the Standard. We therefore strictly adhered to the standards and did not carry out SEM of the samples.

-          Although statistical analyses mentioned many times throughout the article, there is no information on how these analyses carried out and also p values of the comments on the results.

With regard to the presentation, we adhered to the specifications of ISO 5833 and DIN 53435 and did not explicitly present their information again in the article. All values ​​determined were within the standard specifications; the standard deviations are given. The values ​​shown in the figures represent mean values ​​of the individual values ​​specified by the standards including SD.

We used independent two-sample t-tests to compare the means of different formulations of Palacos. This method helps determine if there is a statistically significant difference between the means of two independent groups. (Data Collection: Gather means, standard deviations, and sample sizes for each group. Software used :The calculations were performed using Python with the SciPy library, which is widely used for scientific and statistical computing.

-          Only inhibition zones were evaluated in order to determine the efficacy of the Tig released but there is no indication on the determination of the amount of Tig released from the samples.

In order to provide additional information about the amount of Tig released over the time examined, we would have had to carry out additional very expensive elution including method validation for Tig in PMMA. It is known from the literature that around 10% of the amount of active ingredient used in bone cement is released from a PMMA matrix. The rest remains in the cement matrix and cannot elute. An exact calculation of the amount of active ingredient eluted was not carried out and is also methodologically difficult.

Round 2

Reviewer 4 Report

Comments and Suggestions for Authors

The manuscript which is based on the in vitro mechanical and microbiological evaluation of the commercial bone cement (plain and Gentamycin loaded Palacos) co-loaded with Tigecyclin (Tig) for potential use against antibiotic resistant species has been revised and responses to the review provided by the authors. The abstract and materials&methods sections were properly enhanced. Results section was extended to its final form accordingly with reviewer reports. The statement on the significance of the result can be provided as “statement(p=…)” or “statement (p<0.05)” instead of the paragraph used to mentioned. Regarding the response of the authors to the SEM analysis, authors mentioned that ISO and DIN standards  do not require any additional SEM examination of the molded bodies and they are definitely right on their comment. But, it should be noted that in a research article mentioning mechanical investigations on its title would better provide SEM images of the fracture surfaces in order to prove/support the scientific comments/statements. With this point of view, authors could have provided SEM images since they had incorporated an additive to a standard (it is considered to be standard because of the commercial certifications) product, and authors could have shown the homogeneity of the Tig particle sizes and their homogeneous distribution together with the potential void spaces along the matrix.

The authors enhanced the article significantly in accordance with the reviewer comments.

Author Response

Thank you very much for the interesting comments, which we are very happy to receive.

Reviewer 4: The statement on the significance of the result can be provided as “statement(p=…)” or “statement (p<0.05)” instead of the paragraph used to mentioned.

Authors response : The statement of significance has been modified and now provided as “statement (p=…) or “Statement (p<0,05)”.

Please allow us to point out the following: we looked at some other SEM images of the Copal G+C and Copal G+V, both of the powder and of the fracture surfaces of hardened moldings, and discussed them with our specialists.

Two antibiotics were mixed into both cements. In addition to the antibiotics, the powder also contains polymers, an X-ray contrast agent and benzoyl peroxide. We had a very difficult time interpreting the homogeneous distribution of antibiotic particles in the powder on the SEM images. The antibiotic particles are also difficult to tell apart due to their similar surface of BPO powder.

In fact, small cavities can sometimes be observed on the fracture surfaces of the hardened cement. However, it is by no means clear to conclude that these cavities were the cause of the break. Such small cavities can be filled with body fluids in hydrophilic cements, for example. These can have a dampening effect in vivo and even improve mechanical stability. In the SEM, these small cavities are no longer filled and no longer reflect the in vivo situation. It is also not clear whether antibiotic particles on the fracture surface have an influence on mechanical stability, especially since the antibiotic particles on the cement surface dissolve relatively quickly and elute via diffusion. This certainly happens within 24 hours within the storage times prescribed by the ISO and DIN standards in the SBF medium.

From our point of view, such exciting questions deserve a separate investigation in which mechanical tests are carried out on different cements with different admixtures under different storage conditions.